microsystems/systems theory/software

airborne radar, file management, response time, data storage

**Authors for correspondence:**
Haishan Tian
e-mail: tianhaishan@hunnu.edu.cn
Fangfang Ju
e-mail: paper_new_uni@163.com

# Study on the file management method of data storage system for airborne radar

Haishan Tian[1], Fangfang Ju[1], Hongshan Nie[2], Qiong Yang[1], Yuanyu Wu[1] and Shuangjian Li[1]

[1]School of Physics and Electronics, Hunan Normal University, Changsha 410081, People's Republic of China
[2]College of Electrical and Information Engineering, Hunan University, Changsha 410078, People's Republic of China

(iD) HT, 0000-0002-2887-6026

In order to solve the problem that the air-to-ground data transfer rate is much lower than the radar data rate, the onboard system is commonly used for storing the airborne radar data. However, there are two main problems in the data storage using the traditional file management method. The first is that the frequent data updating of the file allocation table (FAT) and the file directory table (FDT) cause a high frequency of address jumps among the discontinuous areas, which leads to a long response time. The second is that the updating frequencies of the FAT, the FDT and the data region are seriously inconsistent, which results in uneven wear of the three areas. To solve these two problems, a file management method, which optimizes the data writing in the three areas of the FAT, the FDT and the data region, is proposed in this study. An actual measurement is carried out on a data storage system of the airborne radar using the proposed file management method. The result shows that the proposed method significantly reduces the updating frequency of FAT and FDT, and achieves the wear levelling of file area and data region.

## 1. Introduction

Airborne radar has a high data rate that can reach hundreds of Mbps or even Gbps in many applications [1–4], which is much higher than the current air-to-ground data transfer rate, so an onboard data storage system is often required by the airborne radar [5,6]. With the advantages of high storage density, strong vibration resistance, low power consumption, fast storage speed,

**Figure 1.** The structure of the onboard data storage system.

low price and so on, the NAND Flash is the most widely used medium in the data storage system [7–10]. It can adapt to the application scenarios of severe aircraft vibration and harsh working environment and can fully meet the requirement of high storage rate [5,11]. Like other data storage systems, the NAND flash-based onboard storage system also needs file system to manage its internal space, so that the data in the system can be read, written, copied and deleted by the user after the storage [12–15].

The onboard data storage system usually adopts the architecture that the processor controls the NAND flash-based storage medium [16–18], as shown in figure 1. The data writing/reading operations of the storage medium are controlled by the processor, and the internal storage space is managed by a built-in flash translation layer (FTL) controller [19–21]. The FTL is usually invisible and unmodifiable, which is somewhat similar to a black box.

With the limited factors such as small size and low power consumption in the onboard data storage system, its processor has relatively few resources of calculations and caches, which greatly increases the difficulty of file management. Traditional file management methods are not suitable for onboard data storage. Take the file allocation table (FAT) file system, which is widely used in the embedded storage, as an example. With one cluster data stored into the system, the data updating of the FAT and the file directory table (FDT) in the file area is also required in addition to the writing operation of the data region. There are two main problems in this traditional method. The first is that the frequent address jumps lead to long response time. This is because the physical spaces of FAT, FDT and data region are usually discontinuous, and the response time of the data writing among discontinuous spaces is long in the NAND flash. The second is that the data updating frequency of the FAT and FDT is much higher than that in the data region, which results in uneven wear and greatly reduces the life of the storage medium.

Because the FTLs of NAND flash products are not visible to the processor, the studies of optimizing the file management are extensively carried out by scholars of all over the world to solve the two main problems in the onboard data storage system [22–25]. The pre-allocation methods are usually adopted in these studies, such as all clusters pre-allocation (ACPA) method [26] and FAT post- and FDT quasiallocation (FPFQA) method [27]. The ACPA method pre-allocates all unused free clusters before the data storage of a file and releases the unused space after the data storage of this file. This method greatly reduces the updating frequency of the FAT and the FDT during the data storage of this file. But it is not suitable for the case where the size of free storage space is much larger than that of a certain file, for example 32 times the file's size, because the contents of the FAT will be frequently pre-allocated and released in this case. FPFQA method pre-allocates all unused clusters in FAT before data storage and releases the unused clusters after data storage. Therefore, there is no FAT updating during the data storage, and the updating frequency of the FDT is significantly reduced. However, this method is only suitable for a certain pattern of data storage and is not suitable for most applications. Therefore, the research on data storage method for airborne radars is carried out in this study. A method for optimizing the file management is proposed, which can greatly reduce the frequency of address jumps and achieve wear levelling of the storage medium.

The remaining sections of this study are organized as follows. Section 2 describes the background of the study. In §3, the file management optimization method is presented to solve the two problems in data storage. In §4, the performance of the proposed method is tested on a data storage system of the airborne radar. Finally, conclusions are drawn in §5.

## 2. Background

The structure of the file system in data storage is shown in figure 2. It mainly includes two mutually backup FATs, FDT and data region, which are used to establish link relationship of the clusters, describe file directory information and actually store the data, respectively. The physical spaces of the

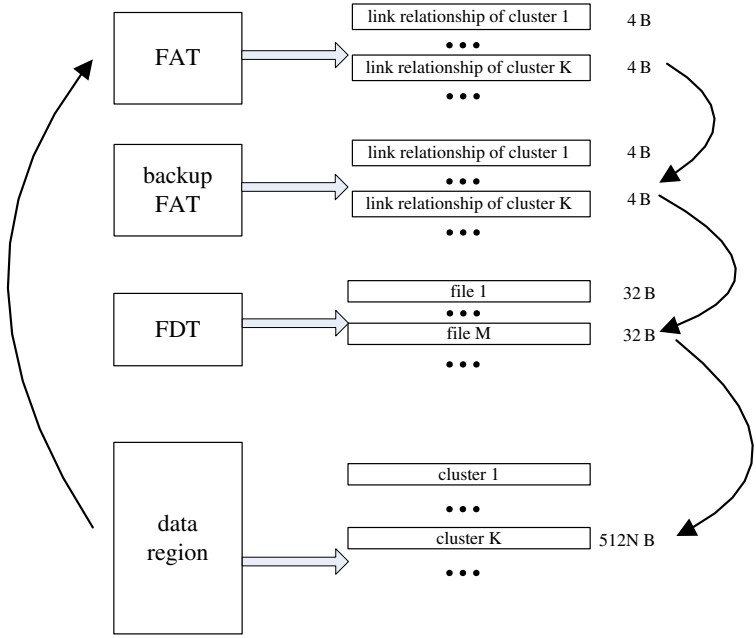

**Figure 2.** The structure of FAT file system.

three parts are discontinuous. The FAT uses the space with the size of 4 B to describe the link relationship of one cluster, and the FDT uses space with the size of 32 B to describe the information of one file. The cluster, which is the smallest unit of file management in a storage system, generally contains dozens of sectors, and the size of one sector is 512 B.

As shown in the arrow of figure 2, there are the data writing operations of three areas in the data storage of one cluster. These three areas are the FATs, the FDT and the data region. So, the data storage time of one cluster is the sum of the data writing time in the three areas, which is introduced as follows.

The data writing time in the data region is

$$T_{DATA\_write} = (512 \times C \times T_w) + T_{w-res},\tag{2.1}$$

where $C$ is the number of sectors included in a cluster, which range from several to tens, depending on the storage medium setting; $T_w$ is the average writing time of 1 B data, depending the writing clock frequency; $T_{w-res}$ is the response time when the data are written into an area which jumps from other discontinuous address. Generally, $T_{w-res}$ is large, maybe several milliseconds (ms) or even dozens of ms, because the original data in an area needs to be erased before the data is written into this area. The value of $T_{w-res}$ is also related to the FTL algorithm of NAND flash product and is generally comparable to the actual data writing time of a single cluster.

Although only the data with the size of 4 B need to be written into FAT, the data of one sector are updated each time because the sector is the smallest unit in the data writing of the NAND flash. So, the frequency of the data writing in FAT is $512/4 = 128$ times that in the data region. The data updating time of the two FATs is

$$T_{FAT\_write} = 2(512 \times T_w + T_{w-res}) = 1024T_w + 2T_{w-res}.\tag{2.2}$$

Because $T_{w-res}$ is much larger than the actual data writing time of one sector $512T_w$, $T_{FAT\_write}$ is approximately equal to $2T_{w-res}$.

Similarly, the data of one sector also need to be written into FDT, which only needs to update 32 B data. The data updating time of the FDT is

$$T_{FDT\_write} = 512 \times T_w + T_{w-res} \approx T_{w-res}.\tag{2.3}$$

The number of files contained in one sector of the FDT is 512 B/32 B = 16. Since the FDT needs to update the sector where the file is located whenever the data of one cluster are stored, the data updating frequency of the FDT is equal to the number of clusters occupied by 16 files.

The data writing time and the wear number of the three areas in the data storage are shown in table 1.

**Table 1.** Data writing time and wear number of the three areas in the FAT file system.

| | data region | FAT | FDT |
|---|---|---|---|
| data writing time per cluster | $512 \cdot C \cdot T_w + T_{w-res}$ | $2T_{w-res}$ | $T_{w-res}$ |
| wear number | 1 | 128 | number of clusters occupied by 16 files |

It can be seen from the table 1 that the sum of the response time caused by address jumps of the three areas is $4T_{w-res}$, which is even much larger than the actual data writing time of one cluster in the data region $512 \cdot C \cdot T_w$. This would significantly reduce the data storage rate. The wear numbers in the FAT and the FDT are 128 times and (the number of clusters occupied by 16 files) times that in the data region, respectively. So, the wear in the three areas is severely uneven, which results in a decrease in the life of the storage medium.

The FAT file management system was originally designed for disks, not for storage systems based on NAND flash. When storing data into storage devices on the ground systems, such as computers, these systems have rich processing and cache resources, which provide powerful hardware resource support for solving the problems and realizing high-speed data storage. However, in the storage system of airborne radar, especially the micro-UAV platform [28,29], the volume, weight and power consumption of the storage system are strictly limited, and computing and cache resources are relatively tight [30,31]. Therefore, higher requirements are put forward for the optimization of the file management of the airborne radar's storage system.

# 3. Improved method of file management

To solve the two problems in the traditional FAT file system, a file management method, which optimizes the data writing of the FAT, the FDT and the data region, is proposed in this study, as shown in figure 3. The following is the introduction of the method.

Step 1: The data of 128 clusters are continuously written into the data region. For example, as shown in the dotted-line box in the upper left corner of figure 3, the data are sequentially written into the data region from cluster N1 + 1 to N1 + 128. In the step 1, only the writing operation of the data region is performed, without the address jumps of the FAT and the FDT and the file data writing.

Step 2: The cluster link information of the 128 clusters in Step 1 is updated in FAT. For example, as shown in the dotted-line box in the upper left corner of figure 3, the link information from cluster N1 + 1 to N1 + 128 is written into the FAT. Because the space with the size of 4 B is used to describe the link relationship of one cluster in FAT, and the sector is the smallest writing unit of the NAND flash, the number of the consecutively written clusters in the data region is set to 512/4 = 128.

Step 3: Repeat the operations of the first two steps until all the data of one file are stored. The operations of the first three steps are as shown in the solid box of figure 3, the data storage of one file is realized by the operations of multiple dotted boxes in the solid box.

The FAT is only updated once whenever the data of 128 clusters are stored through the operation method of the first three steps, which improves the FAT file system in two aspects. The first is that the frequency of address jumps caused by the updating of the FAT is reduced to 1/128 that in the traditional file system. The other is that the data in FAT and in data region are all only updated once during the storage, so that the data region and the FAT are worn out evenly.

Step 4: The FDT information of this file is cached. For example, as shown in the right side of the uppermost solid-line box in figure 3, the file information with the size of 32 B is cached, and temporarily not written into the FDT. The purpose of the cache is to decrease the updating frequency of the FDT, thereby reducing the address jumps and the wear of the FDT.

Step 5: Repeat the operations of the first four steps to realize the data storage of 16 files in the data region and the FAT, and the cache of the FDT information of each file. As shown in figure 3, the operation of the fifth step consists of multiple solid-line boxes which include the data writing in the data region and FAT, and the file information cache.

Step 6: The file information of the 16 files is updated in FDT. As shown in the bottom of figure 3, the cached file information of 16 files is written into the FDT. Because each file's information occupies space with the size of 32 B, and the sector is the smallest unit for data writing, so the number of the continuous written files is set to 512 B/32 B = 16.

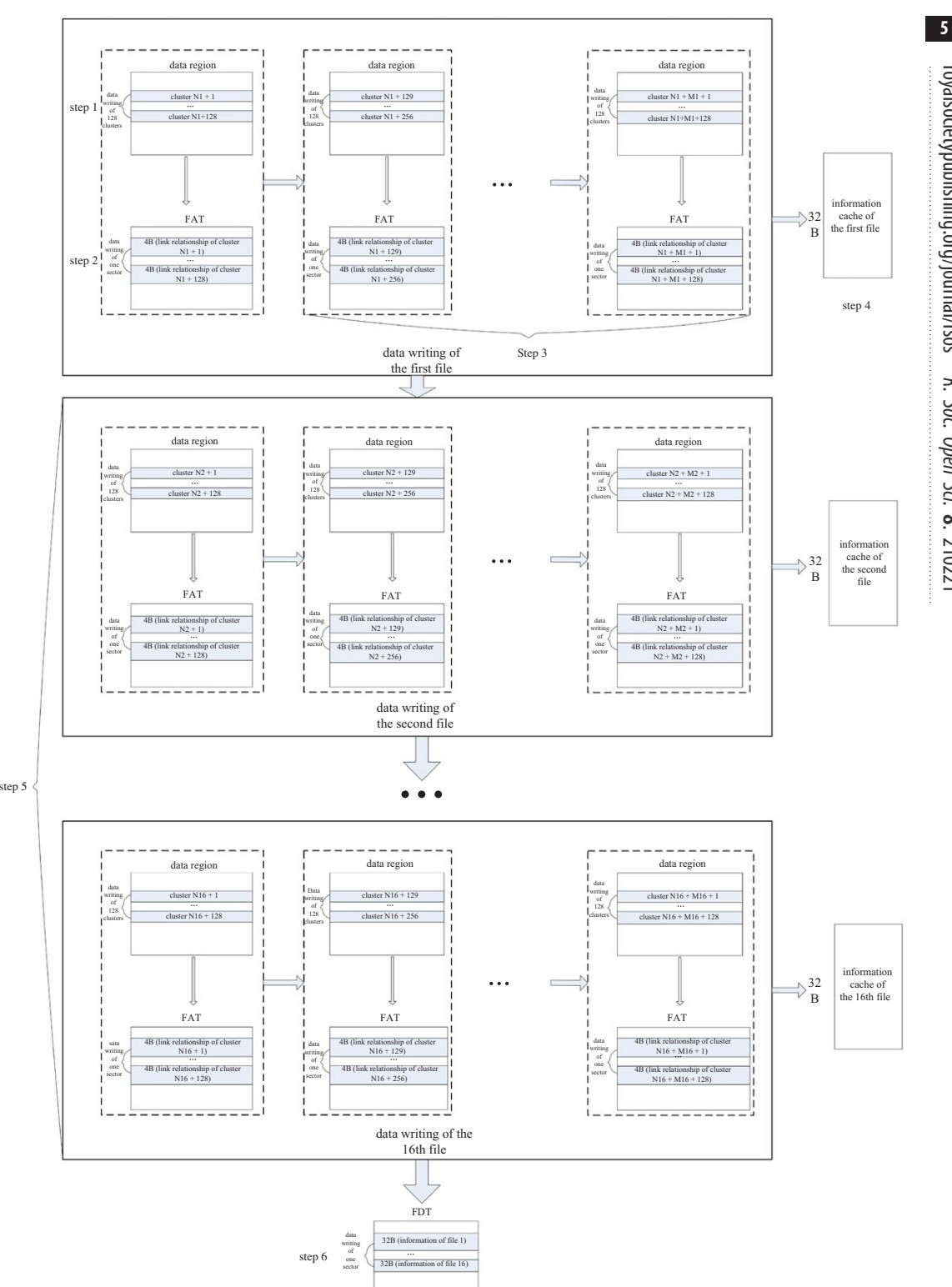

**Figure 3.** Flow chart of the proposed method of file management in data storage for the airborne radar.

Step 7: Follow the operations of the first six steps to realize the subsequent data storage. If there are less than 16 files at the end of the data storage, the file information of the stored files is written into the FDT.

Using the above-proposed file management method, the FAT is only updated once whenever the data of 128 clusters are stored, so the frequency of the address jump caused by the FAT updating is 1/128 that in the traditional FAT system. During the data storage of 16 files, the file information in FDT is only

**Table 2.** Data writing time and wear number of the three areas using ACPA, FPFQA and our proposed method.

| | | ACPA | FPFQA | our proposed method |
|---|---|---|---|---|
| data writing time per cluster | FAT | $8T_w \cdot$ (size of unused space)/ (the number of clusters occupied by 1 file) | $8T_w$ | $2T_{w\text{-}res}/128 = T_{w\text{-}res}/64$ |
| | FDT | $2T_{w\text{-}res}/$(the number of clusters occupied by 1 file) | $2T_{w\text{-}res}/$(the number of clusters occupied by 16 files) | $T_{w\text{-}res}/$(the number of clusters occupied by 16 files) |
| | data region | $512 \cdot C \cdot T_w$ | $512 \cdot C \cdot T_w$ | $512 \cdot C \cdot T_w + T_{w\text{-}res}/128$ |
| wear number | FAT | number of files in unused space | 2 | 1 |
| | FDT | 16 | 1 | 1 |
| | data region | 1 | 1 | 1 |

updated once, so the frequency of the address jump caused by the FDT updating is 1/(the number of clusters occupied by 16 files) that in the FAT system. Combined with the data in table 1, the data writing time and wear number of each area using three methods can be obtained, as shown in table 2.

It can be seen from table 2 that the data region, the FAT and the FDT have the same updating frequency, which achieves wear levelling. The time other than the actual data writing time in table 2 is called the average response time per cluster in this study, which is equal to

$$\frac{T_{w-res}}{128} + \frac{T_{w-res}}{64} + \frac{T_{w-res}}{\text{the number of clusters occupied by 16 files}} \approx \frac{3T_{w-res}}{128}. \tag{3.1}$$

The value of the average response time per cluster in the traditional FAT system is $4T_{w-res}$, as shown in table 1. So, the ratio of this value in the proposed method to that in the traditional system is only

$$\frac{3T_{w-res}/128}{4T_{w-res}} = 0.586\%. \tag{3.2}$$

The value of the average response time per cluster in the ACPA method is approximately equal to $8T_w$ (size of unused space) /(the number of clusters occupied by 1 file), which is much larger than that in the proposed method when the size of unused space exceeds 32 times that of the file. The wear number of FAT and FDT is at least 16 times that of the data region [26].

The value of the average response time per cluster in the FPFQA method is approximately equal to $T_{w\text{-}res}/$(the number of clusters occupied by 16 files), which is smaller than that in the proposed method. However, it is only suitable for a certain pattern that the file size of all data has been determined before storage, and cannot be changed during the storage. Besides that, the wear number of the FAT is twice that of the FDT and data region [27].

Therefore, the file management method proposed in this study could greatly reduce response time caused by the frequent updating of the FAT and FDT in the traditional FAT system and realizes the wear levelling in each area. What is more, the proposed method only adds the cache with the size of $32\,\text{B} \times 16 = 512\,\text{B}$ for the buffer of 16 files' information, and the computing resources required for file management are almost the same as that in the original FAT file system. Therefore, the improved method proposed in this study is suitable for high-speed data storage of airborne radar which has miniature requirement.

## 4. Experimental result

The file management method proposed in this study has been tested on a data storage system of the airborne radar as shown in figure 4. The radar is installed on the aircraft shown in the left figure, and the onboard data storage system is shown in the right figure. FPGA is used as the processor of the

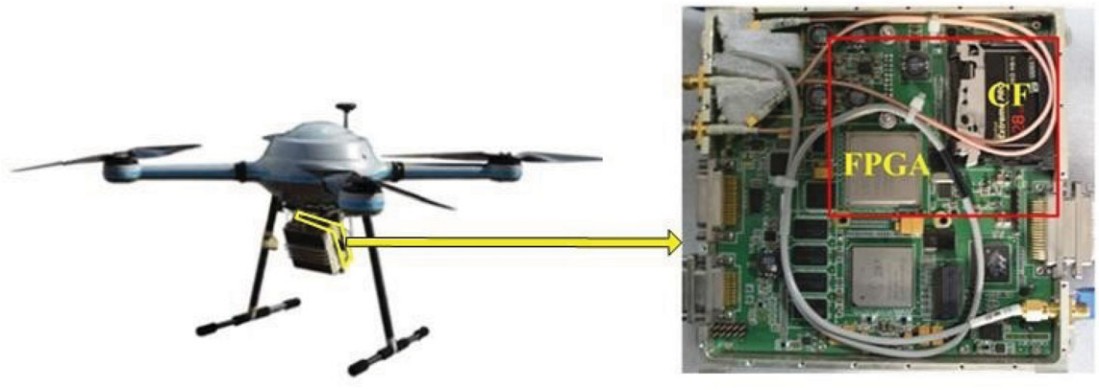

**Figure 4.** The onboard data storage system of an airborne radar.

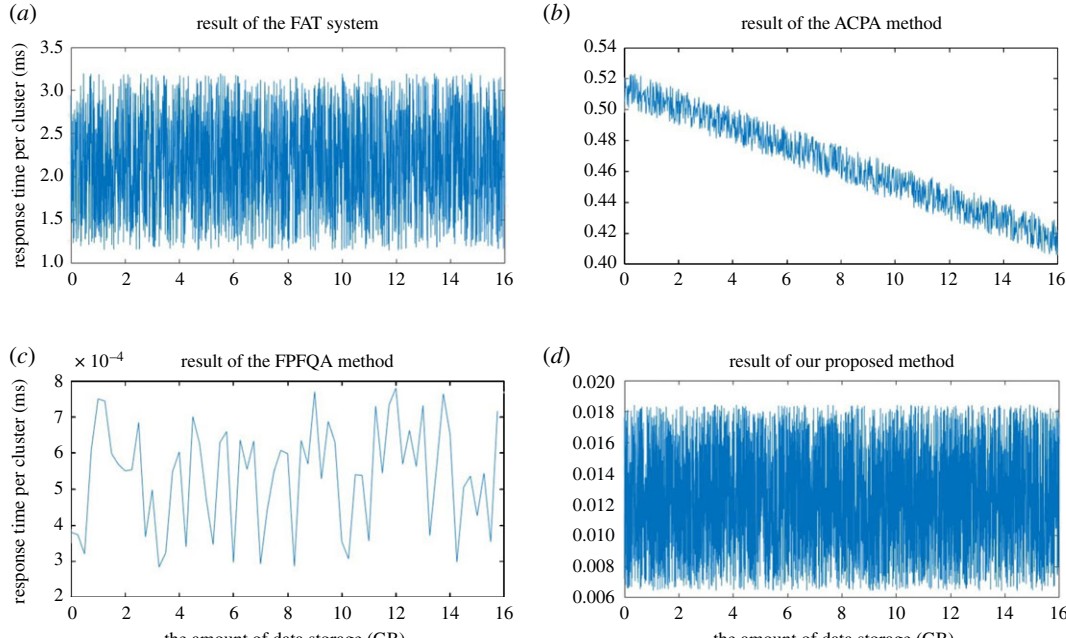

**Figure 5.** Test results of the four file management methods.

storage medium CF card, and the CF card is a NAND flash-based product with the fastest reading and writing speed of 160 MB s$^{-1}$.

In order to verify the performance of the proposed method, the called average response time per cluster and the wear numbers in different areas are tested during the data storage. To compare with other typical improved methods of file management, the traditional FAT file system, ACPA method and FPFQA method are also tested on the data storage system shown in figure 4. In the test of the four methods, the size of each cluster in the CF card is set to 32 kB, and the size of each file is designed to be 16 MB. The data of 1024 files, the size of which is 16 GB, are stored in the CF card in the test. The total number of clusters is $1024 \times 16$ MB/32 kB = 512 k. The test data are linearly increasing data, which are generated by the FPGA and then stored into the CF card, as shown on the right of figure 4. The file management response time of the four methods is measured in the FPGA and exported to the computer through the Chipscope online debugging software. The measured results of the four file management methods are shown in figure 5, where the horizontal axis is the amount of the stored data, and the vertical axis is the average response time per cluster during the storage.

It can be seen from the results in four figures that the response time has some randomness within a range, which is determined by the characteristics of the NAND flash. The measured response time per cluster in the ACPA method shows a decreasing trend, because it is proportional to the size of the unused space which decreased linearly in the test. The sampling points of the FPFQA method are much less than that of other methods. This is because the test result of response time in the FPFQA

**Table 3.** The average response time and the wear number in four methods.

|  |  | FAT | ACPA | FPFQA | our proposed method |
|---|---|---|---|---|---|
| average response time |  | 2.17 ms | 0.465 ms | $5.37 \times 10^{-4}$ ms | 0.0123 ms |
| wear number | FAT | 128 | 512 | 2 | 1 |
|  | FDT | 8 k | 16 | 1 | 1 |
|  | data region | 1 | 1 | 1 | 1 |

method is obtained only after every 16 files are stored, not after each or a certain number of clusters' data are updated. The response time of this method is mainly caused by the data updating in file areas, and the file areas in the FPFQA method are only updated when all 16 files' data are stored in data region.

The response time of the method proposed in this study is much lower than those of the traditional FAT file system and ACPA method, and it is dozens of times that of the FPFQA method. However, the mostly distributed value around 0.012 ms in the proposed method is much smaller than the actual data writing time of one cluster data 32 kB/160 MB s$^{-1}$ = 0.2 ms. Table 3 gives the average value of the results shown in figure 5 and the wear number in the FAT, the FDT and the data region.

As shown in table 3, the ratio of the average response time in our proposed method to that in the traditional FAT system is 0.0123 ms/2.17 ms = 0.567%.

This value approximates the ideal value of 0.586% in formula (3.2). Although the response time of our proposed method is dozens of times higher than that of FPFQA method, the ratio of which to the actual data writing time is only 0.0123 ms/0.2 ms = 6.15%. So, it has a slight impact on the data storage rate. At the same time, the wear number of the FAT in the FPFQA method is twice that of the data region, and this method is only suitable for application where the data is all fixed. Therefore, the experimental results show that the method proposed in this study could greatly reduce the response time, and realize the wear levelling of the FAT, the FDT and the data region. The proposed method is applicable to data storage systems with various capacities, such as the system with the size of several TBs.

# 5. Conclusion

The file management method is studied for the data storage of the airborne radar in this paper. The problems in the traditional file management method are analysed. The updating frequency of the FAT and FDT is very high, which results in frequent address jumps during data storage. Besides this problem, the FAT, FDT and data region wear out unbalanced, which leads to a decrease in the life of the storage medium. To solve the problems in data storage, this study combines the characteristics of the NAND flash and proposes an improved method of file management suitable for the data storage of the airborne radars, and tests it on a real storage system. The experimental results show that the method proposed in this study significantly reduces the updating frequency of the FAT and the FDT, which greatly decrease the address jump frequency, and achieves wear levelling that ensures the service life of the storage medium.

There will be some research to be carried out in the future to further improve storage performance. The following are two possible methods for the study.

— New types of memory, such as phase-change memory that can be replaced by bytes, could be combined with the current widely used NAND Flash to optimize file management and classified store different types of data.
— Software-defined methods can be adopted to develop data storage systems. This method need not change the hardware system, and use the method of the dynamic reconstruction of software and hardware to realize data storage.

Data accessibility. The collected experimental data are made publicly available by the authors. All materials, code and data are provided as electronic supplementary material [32].

Authors' contributions. H.T. made substantial contributions to the conception and design of the study and collected, analysed and interpreted the data. H.N and F.J. carried out the validation of the improved method and the drawing of figures and critically revised the manuscript. Q.Y. and Y.W. coordinated the study and helped draft the

manuscript. All authors gave final approval for publication and agree to be held accountable for the work performed therein.

Competing interests. We declare we have no competing interests.

Funding. This research was funded by Doctoral Science Foundation of Hunan Normal University grant no. 0531120, and General Project of Hunan Provincial Department of Education grant no. 19C1165.

Acknowledgements. The authors would like to thank the anonymous reviewers and editors for their valuable suggestions.

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
