## [Peer Review File · Royal Society Open Science]

Review History

RSOS-210221.R0 (Original submission)

Review form: Reviewer 1

Is the manuscript scientifically sound in its present form?

Yes

Are the interpretations and conclusions justified by the results?

Yes

Is the language acceptable?

Yes

Do you have any ethical concerns with this paper?

No

Have you any concerns about statistical analyses in this paper?

Yes

Recommendation?

Accept with minor revision (please list in comments)

Comments to the Author(s)

The paper proposed an improved file management method of data storage for onboard systems in which real-time high-speed data transfer are not feasible. Comparing to the traditional approach, the improved method showed advantages with respect to the responses time and the address jump frequency. Here are some comments and questions which should be considered before the publication:

- (1) in the third paragraph of Section 1 introduction, "Take the FAT file system, which is widely used in the embedded storage, as an example. With one cluster data stored into the system, the data updating of the file allocation table (FAT) and the file directory table (FDT) in..." , the abbreviation of FAT can only be used after the given full form, not before. Please correct and check all other abbreviations through the paper.
- (2) in the following paragraph, the authors stated that "But it is not suitable for the case where the size of free storage space is much larger than that of a certain file,..". "much larger" is too colloquial. Please give an exact number for example.
- (2) in Section 1 Background, please give a clear definition of the term "cluster".
- (3) in Section 2 and Section 3, it would recommended to assign the variables, such as C, T, with absolute values, giving readers better feeling about the problem.
- (4) in figure 3, please add Steps 1 - 7 in the flowchart explicitly, making the description of the improved method straightforward.
- (5) in Section 3, please distinguish the improved method with other two approaches, ACPA method and FPQA method. What are the advantages and disadvantages of each approach?
- (6) in Section 4, the description of the measurements is too short. Please provide more details to readers for justification. For example, what does the raw data look like? How does the data acquisition system work? How many clusters does the system have? How was the responses time measured?
- (6) in figure 5, why the results of the ACPA method show a decreasing trend? Would the decreasing of the response time continue for large size of CF card, 1024 GB for example? Why the sampling points of the FPFQA method is much less than other?
- (7) in the end of Section 4, please add comments about the performance of the improved approach for TB data storage systems. Does it still qualified?

Review form: Reviewer 2

Is the manuscript scientifically sound in its present form?

Yes

Are the interpretations and conclusions justified by the results?

Yes

Is the language acceptable?

Yes

Do you have any ethical concerns with this paper?

No

Have you any concerns about statistical analyses in this paper?

No

Recommendation?

Major revision is needed (please make suggestions in comments)

Comments to the Author(s)

Overall, this is an interesting work devoted to the context of data storage and management in airborne radars. The proposed study seems sound and worth publication. Still, before recommending it for acceptance, it is my opinion that the following major comments should be addressed by the authors:

- 1) The statement of contributions should be rephrased so as to improve its effectiveness and better stating the technical challenges tackled by the authors.
- 2) Additionally, it would be very useful having a complementary table which categorizes the related works reviewed along their main distinctive characteristics so as to better position the present study by difference.
- 3) In Secs. 2 and 3, I would warmly recommend the authors to revise the content so as to better stress the common challenges of a data storage system for a radar and the peculiarity of adopting this solution (i.e. is this solution general purpose or does it capitalize the unique peculiarities of airborne radar systems?).
- 4) Sec. 4 - It is not clear whether the collected experimental data are (or will be) made publicly available by the authors. In the latter case, this would greatly foster reproducibility and further advances on the topic (and should be clearly stated within the paper, with a pointer to the data repository).
- 5) Please improve the readability of Fig. 5.
- 6) Conclusion section should be enriched with a paragraph highlighting future directions of research, e.g. the application of the proposed methodology to SAR contexts and tracking contexts
 "On the maximal invariant statistic for adaptive radar detection in partially homogeneous disturbance with persymmetric covariance." IEEE Signal Processing Letters 23.12 (2016): 1830-1834.

 "A hash-tree based approach for a totally distributed track oriented multi hypothesis tracker." 2012 IEEE Aerospace Conference. IEEE, 2012.

Decision letter (RSOS-210221.R0)

Dear Dr tian

The Editors assigned to your paper RSOS-210221 "Study on the File Management Method of Data Storage System for Airborne Radar" have now received comments from reviewers and would like you to revise the paper in accordance with the reviewer comments and any comments from the Editors. Please note this decision does not guarantee eventual acceptance.

We invite you to respond to the comments supplied below and revise your manuscript. Below the referees' and Editors' comments (where applicable) we provide additional requirements.

Final acceptance of your manuscript is dependent on these requirements being met. We provide guidance below to help you prepare your revision.

Please submit your revised manuscript and required files (see below) no later than 21 days from today's (ie 07-Apr-2021) date. Note: the ScholarOne system will 'lock' if submission of the revision is attempted 21 or more days after the deadline. If you do not think you will be able to meet this deadline please contact the editorial office immediately.

on behalf of Dr Chong Li (Associate Editor) and R. Kerry Rowe (Subject Editor)
openscience@royalsociety.org

Associate Editor Comments to Author (Dr Chong Li):

Associate Editor: 1

Comments to the Author:

Dear Authors,

Thank you for submitting your work to Royal Society Open Science for consideration of publishing. Based on the feedback from our fellow reviewers, Major Revision is recommended. Please consider all reviewers' comments carefully and address all points raised by the reviewers in the revised manuscript, especially comparison of ACPA method and FPQA method and Section 4 raised by both reviewers.

Yours sincerely

Associate Editor
Dr. Chong Li

Reviewer comments to Author:

Reviewer: 1

Comments to the Author(s)

The paper proposed an improved file management method of data storage for onboard systems in which real-time high-speed data transfer are not feasible. Comparing to the traditional approach, the improved method showed advantages with respect to the responses time and the

address jump frequency. Here are some comments and questions which should be considered before the publication:

- (1) in the third paragraph of Section 1 introduction, "Take the FAT file system, which is widely used in the embedded storage, as an example. With one cluster data stored into the system, the data updating of the file allocation table (FAT) and the file directory table (FDT) in...", the abbreviation of FAT can only be used after the given full form, not before. Please correct and check all other abbreviations through the paper.
- (2) in the following paragraph, the authors stated that "But it is not suitable for the case where the size of free storage space is much larger than that of a certain file,..". "much larger" is too colloquial. Please give an exact number for example.
- (2) in Section 1 Background, please give a clear definition of the term "cluster".
- (3) in Section 2 and Section 3, it would be recommended to assign the variables, such as C, T, with absolute values, giving readers better feeling about the problem.
- (4) in figure 3, please add Steps 1 - 7 in the flowchart explicitly, making the description of the improved method straightforward.
- (5) in Section 3, please distinguish the improved method with other two approaches, ACPA method and FPQA method. What are the advantages and disadvantages of each approach?
- (6) in Section 4, the description of the measurements is too short. Please provide more details to readers for justification. For example, what does the raw data look like? How does the data acquisition system work? How many clusters does the system have? How was the response time measured?
- (6) in figure 5, why the results of the ACPA method show a decreasing trend? Would the decreasing of the response time continue for large size of CF card, 1024 GB for example? Why the sampling points of the FPQA method is much less than other?
- (7) in the end of Section 4, please add comments about the performance of the improved approach for TB data storage systems. Does it still qualify?

Reviewer: 2

Comments to the Author(s)

Overall, this is an interesting work devoted to the context of data storage and management in airborne radars. The proposed study seems sound and worth publication. Still, before recommending it for acceptance, it is my opinion that the following major comments should be addressed by the authors:

- 1) The statement of contributions should be rephrased so as to improve its effectiveness and better stating the technical challenges tackled by the authors.
- 2) Additionally, it would be very useful having a complementary table which categorizes the related works reviewed along their main distinctive characteristics so as to better position the present study by difference.
- 3) In Secs. 2 and 3, I would warmly recommend the authors to revise the content so as to better stress the common challenges of a data storage system for a radar and the peculiarity of adopting this solution (i.e. is this solution general purpose or does it capitalize the unique peculiarities of airborne radar systems?).
- 4) Sec. 4 - It is not clear whether the collected experimental data are (or will be) made publicly available by the authors. In the latter case, this would greatly foster reproducibility and further advances on the topic (and should be clearly stated within the paper, with a pointer to the data repository).

5) Please improve the readability of Fig. 5.

6) Conclusion section should be enriched with a paragraph highlighting future directions of research, e.g. the application of the proposed methodology to SAR contexts and tracking contexts "On the maximal invariant statistic for adaptive radar detection in partially homogeneous disturbance with persymmetric covariance." IEEE Signal Processing Letters 23.12 (2016): 1830-1834.

"A hash-tree based approach for a totally distributed track oriented multi hypothesis tracker." 2012 IEEE Aerospace Conference. IEEE, 2012.

===PREPARING YOUR MANUSCRIPT===

===PREPARING YOUR REVISION IN SCHOLARONE===

Author's Response to Decision Letter for (RSOS-210221.R0)

See Appendices A - C.

RSOS-210221.R1 (Revision)

Review form: Reviewer 1

Is the manuscript scientifically sound in its present form?

Yes

Are the interpretations and conclusions justified by the results?

Yes

Is the language acceptable?

Yes

Do you have any ethical concerns with this paper?

No

Have you any concerns about statistical analyses in this paper?

No

Recommendation?

Accept with minor revision (please list in comments)

Comments to the Author(s)

Thanks to all authors for the effort. All the comments have been clearly clarified. In order to improve the readability of all Tables and Figures, please adjust the size of the fonts. Some text are too small and some numbers are too large. Please also replot Figure 5 into four subfigures with the titles into the caption as (a),(b),(c),(d).

Review form: Reviewer 2

Is the manuscript scientifically sound in its present form?

Yes

Are the interpretations and conclusions justified by the results?

Yes

Is the language acceptable?

Yes

Do you have any ethical concerns with this paper?

No

Have you any concerns about statistical analyses in this paper?

No

Recommendation?

Accept as is

Comments to the Author(s)

Overall, this is an interesting work devoted to the context of data storage and management in airborne radars.

The authors have satisfactorily addressed my previous comments and modified their manuscript accordingly. Hence, I am glad to recommend the present work for publication.

Decision letter (RSOS-210221.R1)

Dear Dr tian,

It is a pleasure to accept your manuscript entitled "Study on the File Management Method of Data Storage System for Airborne Radar" in its current form for publication in Royal Society Open Science. The comments of the reviewer(s) who reviewed your manuscript are included at the foot of this letter.

on behalf of Dr Chong Li (Associate Editor) and R. Kerry Rowe (Subject Editor)
openscience@royalsociety.org

Reviewer comments to Author:

Reviewer: 2

Comments to the Author(s)

Overall, this is an interesting work devoted to the context of data storage and management in airborne radars.

The authors have satisfactorily addressed my previous comments and modified their manuscript accordingly. Hence, I am glad to recommend the present work for publication.

Reviewer: 1

Comments to the Author(s)

Thanks to all authors for the effort. All the comments have been clearly clarified. In order to improve the readability of all Tables and Figures, please adjust the size of the fonts. Some text are too small and some numbers are too large. Please also replot Figure 5 into four subfigures with the titles into the caption as (a),(b),(c),(d).

Appendix A

Dear Dr. Prof. Chong Li,

Thank you for sending us the reviewer's comments. Please find the revised manuscript, which we would like to submit for consideration of publishing.

Title: "Study on the File Management Method of Data Storage System for Airborne Radar".

Authors: Haishan Tian, Fangfang Ju, Hongshan Nie, Qiong Yang, Yuanyu Wu, Shuangjian Li.

We carried out the required revisions carefully, especially the comparison of ACPA method, FPQA method and the improved method, which is raised by both reviewers. The modifications are listed in the **Summary of the changes**. We also attached **Response to the Reviewer** as a separated sheet.

We hope the revised MS will be satisfactory for publication in your journal. Thank you for your kind efforts and look forward to your early response.

Yours sincerely,

Dr. Haishan Tian

School of Physics and Electronics
Hunan Normal University
Changsha 410081, People's Republic of China

Summary of the changes in manuscript

- (1) Following the reviewer's suggestion, we have modified the sentence to correct the imprecise statement for the abbreviation of FAT "Take the **file allocation table (FAT)** file system, ..., the data updating of the **FAT** and ...". (**paragraph 3 of Section 1**).
- (2) We have added an exact number for the phrasing "much larger" to avoid the possible confusion "..., **for example 32 times the file's size, ...**". (**paragraph 4 of Section 1**).
- (3) we have added a clear definition of the term cluster "The cluster, **which is the smallest unit of file management in a storage system, ...**". (**paragraph 1 of Section 2**).
- (4) Following the reviewer's suggestion, we have assigned the variables, such as C , T_w , T_{w-res} with absolute values or value range "where C is ..., **which range is from several to tens, ...**; T_w is ..., **depending the writing clock frequency**; T_{w-res} is....

Generally, T_{w-res} is ..., maybe several milliseconds (ms) or even dozens of ms, ...”.

(paragraph 3 of Section 2).

(5) Following the reviewer’s suggestion, we have added one paragraph to better stress the common challenges of a data storage system for a radar and the peculiarity of adopting this solution “The FAT file management system was originally designed for disks, ... Therefore, higher requirements are put forward for the optimization of the file management of the airborne radar’s storage system.”. (paragraph 10 of Section 2) and add Ref.31, 32.

(6) Following the reviewer’s suggestion, we have enriched the table 2 which categorizes the related works reviewed along their main distinctive characteristics to better position the present study by difference. (Table 2).

Table 2 Data writing time and wear number of the three areas using ACPA, FPFQA and the proposed method

		ACPA	FPFQA	Our proposed method
data writing time per cluster	FAT	$8T_w \cdot$ (size of unused space) / (the number of clusters occupied by 1 file)	$8 T_w$	$2T_{w-res} / 128 = T_{w-res} / 64$
	FDT	$2T_{w-res}$ / (the number of clusters occupied by 1 file)	T_{w-res} / (the number of clusters occupied by 16 files)	T_{w-res} / (the number of clusters occupied by 16 files)
	Data region	$512 \cdot C \cdot T_w$	$512 \cdot C \cdot T_w$	$512 \cdot C \cdot T_w + T_{w-res} / 128$
Wear number	FAT	number of files in unused space	2	1
	FDT	16	1	1
	Data region	1	1	1

(7) We have added two paragraphs to distinguish the improved method with other two approaches, ACPA method and FPQA method. (paragraph 13 and 14 of Section 3).

The value of the average response time per cluster in the ACPA method ... The wear number of FAT and FDT is at least 16 times that of the data region [29].

The value of the average response time per cluster in the FPFQA method ... the wear number of the FAT is twice that of the FDT and data region [30].

(8) We have added sentences to compare the used computing and cache resources in the improved method with that in the original FAT file system. “What's more, the proposed method only adds ... suitable for high-speed data storage of airborne radar which have miniature requirement.”. (paragraph 15 of Section 3).

(9) We have added Steps 1 - 6 in the flowchart explicitly (New Figure 3).

New Figure 3. Flow chart of the proposed method of file management in data storage for the airborne radar

- (10) Following the reviewer's suggestion, we have added more details for the description of the measurements in Section 4 "The total number of clusters ... through the Chipscope online debugging software." (**paragraph 2 of Section 4**).
- (11) Following the reviewer's suggestion, we have added the sentences to describe and analyze the results of Fig.5 "The measured response time per cluster in the ACPA method ... the FPFQA method is only updated when all 16 files' data are stored into data region." (**paragraph 3 of Section 4**).
- (12) We have added comments about the performance of the improved approach for TB data storage systems "The proposed method is applicable to data storage systems with various capacities, such as the system with the size of several TBs." (**paragraph 6 of Section 4**).
- (13) Following the reviewer's suggestion, we have enriched conclusion section by adding parts highlighting future directions of research. (**paragraph 2, 3 and 4 of Section 5**).
- There will be some research ... two possible methods for the study.
- New types of memory ... classified store different types of data;
 - Software-defined methods ... realize data storage.
- (14) We have made it clear that the collected experimental data will be made publicly available "The collected experimental data are made publicly available by the authors. All materials, code and data are contained in the submitted supplementary material". (**Data Accessibility**).
- (15) Following the reviewer's suggestion, we have rephrased statement of contributions to improve its effectiveness and better stating the technical challenges tackled by the authors "H.T. made ... accountable for the work performed therein". (**Authors' Contributions**).

Appendix B

Response to the Reviewer 1

Comment:

The paper proposed an improved file management method of data storage for onboard systems in which real-time high-speed data transfer are not feasible. Comparing to the traditional approach, the improved method showed advantages with respect to the responses time and the address jump frequency. Here are some comments and questions which should be considered before the publication:

Response:

We sincerely thank the reviewer for careful reading of the manuscript and for his/her positive review. We are delighted to address the items raised by the reviewer.

Comment:

in the third paragraph of Section 1 introduction, “Take the FAT file system, which is widely used in the embedded storage, as an example. With one cluster data stored into the system, the data updating of the file allocation table (FAT) and the file directory table (FDT) in...””, the abbreviation of FAT can only be used after the given full form, not before. Please correct and check all other abbreviations through the paper.

Response:

We thank the reviewer for raising this important point, which provides us the opportunity to correct the imprecise statements. Following the reviewer’s suggestion, we have modified the sentence in **paragraph 3 of Section 1**: “Take the **file allocation table (FAT)** file system, which is widely used in the embedded storage, as an example. With one cluster data stored into the system, the data updating of the **FAT** and the file directory table (FDT) in...”. All other abbreviations are checked through the paper.

Comment:

in the following paragraph, the authors stated that “But it is not suitable for the case where the size of free storage space is much larger than that of a certain file,..”. “much larger” is too colloquial. Please give an exact number for example.

Response:

We thank the reviewer for raising this important point. Following the reviewer’s suggestion, we have added an exact number for example in **paragraph 4 of Section 1**: “..., **for example 32 times the file’s size, ...”.**

Comment:

in Section 1 Background, please give a clear definition of the term “cluster”.

Response:

The reviewer has raised a valuable point that needs to be clarified. Following the reviewer's suggestion, we have added a clear definition of the term "cluster" in **paragraph 1 of Section 2 Background**: "The cluster, which is the smallest unit of file management in a storage system, ..."

Comment:

in Section 2 and Section 3, it would recommended to assign the variables, such as C, T, with absolute values, giving readers better feeling about the problem.

Response:

We thank the reviewer for raising this important point. Following the reviewer's suggestion, we have assigned the variables, such as C, T_w , T_{w-res} with absolute values or value range in **paragraph 3 of Section 2**: "where C is the number of sectors included in a cluster, which range is from several to tens, depending on the storage medium setting; T_w is the average writing time of 1B data, depending the writing clock frequency; T_{w-res} is the response time when the data are written into an area which jumps from other discontinuous address. Generally, T_{w-res} is large, maybe several milliseconds (ms) or even dozens of ms ..."

Comment:

in figure 3, please add Steps 1 - 7 in the flowchart explicitly, making the description of the improved method straightforward.

Response:

We thank the reviewer for raising this important point, which provides us an opportunity to making the description of the improved method straightforward. Following the reviewer's suggestion, we have added Steps 1 - 6 in the flowchart explicitly [see **New Figure 3**]. The step 7 is not added, because this step repeats the operations of Steps 1 - 6, and is not easy to identify in the figure.

New Figure 3. Flow chart of the proposed method of file management in data storage for the airborne radar

Comment:

in Section 3, please distinguish the improved method with other two approaches, ACPA method and FPQA method. What are the advantages and disadvantages of each approach?

Response:

We thank the reviewer for raising this important point, which provides us an opportunity to clarify the problem and avoid the possible confusion. Following the reviewer's suggestion, we have added **paragraph 13 and 14 of Section 3** to distinguish the improved method with other two approaches, ACPA method and FPQA method:

The value of the average response time per cluster in the ACPA method is approximately equal to $8T_w \times (\text{size of unused space}) / (\text{the number of clusters occupied by 1 file})$, which is much larger than that in proposed method when the size of unused space exceeds 32 times that of the file. The wear number of FAT and FDT is at least 16 times that of the data region [29].

The value of the average response time per cluster in the FPFQA method is approximately equal to $2 T_{w-res} / (\text{the number of clusters occupied by 16 files})$, which is smaller than that in proposed method. However, it is only suitable for a certain pattern that the file size of all data has been determined before storage, and can't be changed during the storage. Besides that, the wear number of the FAT is twice that of the FDT and data region [30].

Comment:

in Section 4, the description of the measurements is too short. Please provide more details to readers for justification. For example, what does the raw data look like? How does the data acquisition system work? How many clusters does the system have? How was the responses time measured?

Response:

We thank the reviewer for raising this important point, which provides us an opportunity to describe the measurements clearly. Following the reviewer's suggestion, we have added more details for the description of the measurements on **paragraph 2 of Section 4**: "The total number of clusters are $1024 \times 16\text{MB} / 32\text{kB} = 512\text{k}$. The test data are linearly increasing data, which are generated by the FPGA and then stored into the CF card, as shown on the right of Figure 4. The file management response time of the four methods is measured in the FPGA and exported to the computer through the Chipscope online debugging software."

Comment:

in figure 5, why the results of the ACPA method show a decreasing trend? Would the decreasing of the response time continue for large size of CF card, 1024 GB for example? Why the sampling points of the FPFQA method is much less than other?

Response:

We sincerely thank the reviewer for raising these important points that needs to be clarified. The results of the ACPA method show a decreasing trend, because the response time is proportional to the size of the unused space which decreased linearly in the test. The reduction in response time is related to the decrease in the size of the unused space of the storage medium, and has nothing to do with the size of CF card.

The sampling points of the FPFQA method is much less than that of other methods. This is because the test result of response time in the FPFQA method is obtained only after every 16 files are stored, not after each or a certain number of clusters' data are updated. The response time of this method is mainly caused by the data updating in file areas, and the file areas in the FPFQA method is only updated when all 16 files' data are stored into data region.

To clarify the problem and avoids the possible confusion we have added the sentences in **paragraph 3 of Section 4**: “The measured response time per cluster in the ACPA method shows a decreasing trend, because it is proportional to the size of the unused space which decreased linearly in the test. The sampling points of the FPFQA method is much less than that of other methods. This is because the test result of response time in the FPFQA method is obtained only after every 16 files are stored, not after each or a certain number of clusters' data are updated. The response time of this method is mainly caused by the data updating in file areas, and the file areas in the FPFQA method is only updated when all 16 files' data are stored into data region.”.

Comment:

in the end of Section 4, please add comments about the performance of the improved approach for TB data storage systems. Does it still qualified?

Response:

We thank the reviewer for raising this important issue, which provides us an opportunity to clarify the problem. The performance of the improved approach in TB data storage systems is same with that in the test system. It still qualifies. Following the reviewer's suggestion, we have added comments about the performance of the improved approach for TB data storage systems in **paragraph 6 of Section 4**: “The proposed method is applicable to data storage systems with various capacities, such as the system with the size of several TBs.”.

Appendix C

Response to the Reviewer 2

Comment:

Overall, this is an interesting work devoted to the context of data storage and management in airborne radars. The proposed study seems sound and worth publication. Still, before recommending it for acceptance, it is my opinion that the following major comments should be addressed by the authors:

Response:

We sincerely thank the reviewer for carefully reading our manuscript and raising these important points, which have helped us a lot in improving our manuscript clarity. We are delighted to address the items raised by the reviewer.

Comment:

The statement of contributions should be rephrased so as to improve its effectiveness and better stating the technical challenges tackled by the authors.

Response:

We thank the reviewer for pointing out this important issue. Following the reviewer's suggestion, we have rephrased statement of contributions to improve its effectiveness and better stating the technical challenges tackled by the authors on **Authors' Contributions**: "H.T. made substantial contributions to conception and design of the study and collected, analyzed, and interpreted the data. H.N and F.J. carried out the validation of the improved method and the drawing of figures and critically revised the manuscript. Q.Y. and Y.W. coordinated the study, and helped draft the manuscript. All authors gave final approval for publication and agree to be held accountable for the work performed therein."

Comment:

Additionally, it would be very useful having a complementary table which categorizes the related works reviewed along their main distinctive characteristics so as to better position the present study by difference.

Response:

We thank the reviewer for raising this important point. Following the reviewer's suggestion, we have enriched the **Table 2** which categorizes the related works reviewed along their main distinctive characteristics so as to better position the present study by difference.

Table 2 Data writing time and wear number of the three areas using ACPA, FPFQA and the proposed method

		ACPA	FPFQA	Our proposed method
data writing time per cluster	FAT	$8T_w \cdot (\text{size of unused space}) / (\text{the number of clusters occupied by 1 file})$	$8 T_w$	$2T_{w-res} / 128 = T_{w-res} / 64$
	FDT	$2T_{w-res} / (\text{the number of clusters occupied by 1 file})$	$2T_{w-res} / (\text{the number of clusters occupied by 16 files})$	$T_{w-res} / (\text{the number of clusters occupied by 16 files})$
	Data region	$512 \cdot C \cdot T_w$	$512 \cdot C \cdot T_w$	$512 \cdot C \cdot T_w + T_{w-res} / 128$
Wear number	FAT	number of files in unused space	2	1
	FDT	16	1	1
	Data region	1	1	1

Besides that, we have added **paragraph 13 and 14 of Section 3** to distinguish the improved method with other two approaches, ACPA method and FPQA method:

The value of the average response time per cluster in the ACPA method is approximately equal to $8T_w \cdot (\text{size of unused space}) / (\text{the number of clusters occupied by 1 file})$, which is much larger than that in proposed method when the size of unused space exceeds 32 times that of the file. The wear number of FAT and FDT is at least 16 times that of the data region [29].

The value of the average response time per cluster in the FPFQA method is almost $2 T_{w-res} \times (\text{the number of clusters occupied by 16 files})$, which is smaller than that in proposed method. However, it is only suitable for a certain pattern that the file size of all data has been determined before storage, and can't be changed during the storage. Besides that, the wear number of the FAT is twice that of the FDT and data region [30].

Comment:

In Secs. 2 and 3, I would warmly recommend the authors to revise the content so as to better stress the common challenges of a data storage system for a radar and the peculiarity of adopting this solution (i.e. is this solution general purpose or does it capitalize the unique peculiarities of airborne radar systems?).

Response:

We sincerely and respectfully thank the reviewer for this warm recommendation. We are delighted to revise the content so as to better stress the common challenges of a data storage system for a radar and the peculiarity of adopting this solution. Following the

reviewer's suggestion, we have added one paragraph to explain that this solution is unique for airborne radar systems due to its limited computing and cache resources under the miniature requirement of the platform. The added part is **paragraph 10 of Section 2:**

The FAT file management system was originally designed for disks, not for storage systems based on NAND flash. When storing data into storage devices on the ground systems, such as computers, these systems have rich processing and cache resources, which provide powerful hardware resource support for solving the problems and realizing high-speed data storage. However, in the storage system of airborne radar, especially the micro-UAV platform [31, 32], the volume, weight, and power consumption of the storage system are strictly limited, and computing and cache resources are relatively tight. Therefore, higher requirements are put forward for the optimization of the file management of the airborne radar's storage system.

Besides that, the used computing and cache resources in the improved method are compared with that in the original FAT file system. The result is that the improved method uses almost the same computing resources and adds only the cache with the size of 512B. So, the improved method is suitable for the storage system of the airborne radar under the miniature requirement. The analysis of the result is in **paragraph 15 of Section 3:** "What's more, the proposed method only adds the cache with the size of $32B \times 16 = 512B$ for the buffer of 16 files' information, and the computing resources required for file management are almost the same as that in the original FAT file system. Therefore, the improved method proposed in this study is suitable for high-speed data storage of airborne radar which have miniature requirement."

Comment:

Sec. 4 – It is not clear whether the collected experimental data are (or will be) made publicly available by the authors. In the latter case, this would greatly foster reproducibility and further advances on the topic (and should be clearly stated within the paper, with a pointer to the data repository).

Response:

We thank the reviewer for pointing out the important issue. The collected experimental data will be made publicly available by the authors. Four excel files, which include the collected experimental data, and one code file are contained in the supplementary material. we have made it clear that the collected experimental data will be made publicly available on **Data Accessibility:** "The collected experimental data are made publicly available by the authors. All materials, code and data are provided as electronic supplementary material."

Comment:

Please improve the readability of Fig. 5.

Response:

Following the reviewer's suggestion, we have added the sentences to describe and analyze the results of Fig.5 in **paragraph 3 of Section 4**: "The measured response time per cluster in the ACPA method shows a decreasing trend, because it is proportional to the size of the unused space which decreased linearly in the test. The sampling points of the FPFQA method is much less than that of other methods. This is because the test result of response time in the FPFQA method is obtained only after every 16 files are stored, not after each or a certain number of clusters' data are updated. The response time of this method is mainly caused by the data updating in file areas, and the file areas in the FPFQA method is only updated when all 16 files' data are stored into data region.". Besides that, we added more details for the description of the measurements for obtaining the results of Fig.5 in **paragraph 2 of Section 4**: "The total number of clusters are $1024 \times 16\text{MB} / 32\text{kB} = 512\text{k}$. The test data are linearly increasing data, which are generated by the FPGA and then stored into the CF card, as shown on the right of Figure 4. The file management response time of the four methods is measured in the FPGA and exported to the computer through the Chipscope online debugging software."

Comment:

Conclusion section should be enriched with a paragraph highlighting future directions of research, e.g. the application of the proposed methodology to SAR contexts and tracking contexts

"On the maximal invariant statistic for adaptive radar detection in partially homogeneous disturbance with persymmetric covariance." IEEE Signal Processing Letters 23.12 (2016): 1830-1834.

"A hash-tree based approach for a totally distributed track oriented multi hypothesis tracker." 2012 IEEE Aerospace Conference. IEEE, 2012.

Response:

We thank the reviewer for raising this important point. Following the reviewer's suggestion, we have enriched conclusion section by adding parts highlighting future directions of research in **paragraph 2, 3 and 4 of Section 5**:

There will be some research to be carried out in the future to further improve storage performance. The following are two possible methods for the study.

- New types of memory, such as phase change memory that can be replaced by bytes, could be combined with the current widely used NAND Flash to optimize file management and classified store different types of data;
- Software-defined methods can be adopted to develop data storage systems. This method need not change the hardware system, and uses the method of the dynamic reconstruction of software and hardware to realize data storage.